# Inclusion of Older Adults in the Research and Design of Digital Technology

**DOI:** 10.3390/ijerph16193718

**Published:** 2019-10-02

**Authors:** Ittay Mannheim, Ella Schwartz, Wanyu Xi, Sandra C. Buttigieg, Mary McDonnell-Naughton, Eveline J. M. Wouters, Yvonne van Zaalen

**Affiliations:** 1School of Allied Health Professions, Fontys University of Applied Science, Eindhoven 5631 BN, The Netherlands; e.wouters@fontys.nl (E.J.M.W.); y.vanzaalen@fontys.nl (Y.v.Z.); 2Tranzo, School of Social and Behavioral Sciences, Tilburg University, Tilburg 5037 DB, The Netherlands; 3Louis and Gabi Weisfeld School of Social Work, Bar Ilan University, Ramat Gan 5290002, Israel; ella.schwartz1@mail.huji.ac.il (E.S.); wanyu.xi@biu.ac.il (W.X.); 4Department of Public Health, Faculty of Health Sciences, Ben-Gurion University of the Negev, Be’er Sheva 8410501, Israel; 5Health Services Management, Faculty of Health Sciences, University of Malta, Msida MSD 2090, Malta; sandra.buttigieg@um.edu.mt; 6Health Services Management Centre, School of Social Policy, College of Social Sciences, University of Birmingham, Edgbaston, Birmingham B15 2TT, UK; 7Department of Nursing and Health Care Athlone Institute of Technology, Athlone N37 HD68, Ireland; mmcdonnell@ait.ie

**Keywords:** digital technology, older adults, ageism, inclusion, ethics

## Abstract

Digital technology holds a promise to improve older adults’ well-being and promote ageing in place. However, there seems to be a discrepancy between digital technologies that are developed and what older adults actually want and need. Ageing is stereotypically framed as a problem needed to be fixed, and older adults are considered to be frail and incompetent. Not surprisingly, many of the technologies developed for the use of older adults focus on care. The exclusion of older adults from the research and design of digital technology is often based on such negative stereotypes. In this opinion article, we argue that the inclusion rather than exclusion of older adults in the design process and research of digital technology is essential if technology is to fulfill the promise of improving well-being. We emphasize why this is important while also providing guidelines, evidence from the literature, and examples on how to do so. We unequivocally state that designers and researchers should make every effort to ensure the involvement of older adults in the design process and research of digital technology. Based on this paper, we suggest that ageism in the design process of digital technology might play a role as a possible barrier of adopting technology.

## 1. Introduction

Modern times present us with two main challenges and opportunities. First, the ageing of the population and second, the exponential development of digital technology (DT), also known as the digital revolution or the fourth industrial revolution [1]. As life expectancy is being prolonged, the proportion of older adults aged 65 and above is projected to reach 30% by 2050 in many western countries and almost 40% in the world’s leading ageing country, Japan [2]. The dramatic increase of older population has occurred due to several reasons, mainly better health care and improved living conditions but also due to advances in technology [3].

DT influences our lives in various ways. In relation to older adults, it holds a promise to improve well-being [4], healthcare [5], and to support ageing in place in a safe and independent environment [6]. For the purpose of this study, we describe DT as technological devices, services or platforms that use, collect, and often process data and are connected to the internet, other devices, or apps. Examples of such DTs are sensors and systems that can detect and predict falls, eHealth apps, and social robots. Gerontology studies on DT include applications to physical and mental health, mobility, social connectedness, loneliness, communication, leisure and safety [7]. Most of these studies do not focus on technology that is designed together with older persons. It is unknown if and how older adults are involved and considered when designing and researching DT or if they are excluded. Additionally, we do not know if the DTs that are developed meet the actual needs and desires of older adults, or, rather, we do not know what DT designers think older adults need and want. Are these technologies also meant for the use of older adults with cognitive decline? Are some older adults left behind?

As noted by Schultz et al. [7], much of the DT developed today for the use of older adults focuses on care and healthcare. Ageing is framed as posing a challenge or ‘problem’ in which technology is set to intervene and ‘fix’, in what was described by Peine and Neven [8] as an “interventionist logic”. Contrary to this “interventionist logic”, there seems to be a mismatch between the DTs that are developed and what older end-users actually adopt and use [9]. Extensive research has focused on acceptance models (e.g., [10]) and on predictors and barriers to adoption of technology [11,12,13]. However, not much research has focused on what older adults really want, and, furthermore, much research has not taken the heterogeneity of older adults into consideration [9]. When it comes to technology, older adults are often stereotypically described as a homogeneous group that are lagging behind and associated with cognitive decline, frailty and needs. These stereotypes often affect designers in the design process of DT [14]. Though some studies have implied that the adoption of technology among older adults is on the rise [15] and that older adults hold more positive views (than negative) about using technology [16], it is still the case that negative stereotypes and ageism might shape how DT for older adults is conceived and designed [17,18].

Negative stereotypes of older adults as being incompetent may lead to different forms of social exclusion [19]. Such exclusion varies across several domains such as the accessibility to services and resources, social relations, community, and civic participation [20]. Indeed, such exclusion is also present in research, specifically in clinical research and randomized controlled trials [21,22,23,24]. Most importantly, it seems that many studies use upper age limits as an unjustified exclusion criterion, with some estimates showing that more than half of the studies approved by an ethical committee with relevance to older people had an unjustified upper age limit [21]. Though a more recent study might suggest that the use of unjustified upper age limits is decreasing, selective exclusion is still prominent [25]. While many studies did not give any justification of upper age limits, some researchers’ justifications for exclusion often reflected negative ageist views of older people, stressing that “participants need to be ‘fully competent,’ ‘reliable,’ or ‘without cognitive impairment’” [21] (p. 993). This approach demonstrates the ethical dilemmas that researchers face regarding older adults’ capability to participate, provide valuable information, and to knowingly consent. However, this can also reflect how stereotypes of older adults affect design and research. Consequently, research involving older adults with cognitive decline and dementia is even more sparse [26,27]. In a recent literature review by van Boekel et al. [28] on stakeholders’ perspectives on technology use by people with dementia, it was found that the main perspective investigated in studies has been that of the family caregiver. The perspective of the people with dementia has hardly been explored, and when it has been, it has been many times via a proxy. Growing evidence has demonstrated the valuable information that can be obtained from people with dementia and people with disabilities both in research [26,29,30,31,32,33] and in the design of technology [34,35]. In relation to technology, obtaining older adults’ (with or without cognitive decline) perspectives directly is of great importance, as older adults often hold different perspectives than other stakeholders [12,28].

Other possible concerns that researchers and designers hold relate to additional ethical considerations, which is to say potential harm to autonomy, privacy and dignity. In addition, using technology in research and interventions has additional implications to be considered regarding safety and security, trust and equitable resources [36]. The exclusion of older adults and the avoidance of confronting these concerns as a challenge might be the “easy answer” for some researchers and professionals [37] (p. 125), though this approach imposes age-based discrimination.

While the exclusion of older adults from the research and development of technology might affect the quality of data regarding aging populations and limit external validity [38], including a variety of older users may provide a better and wider understanding of the subject at hand. Involving older adults in research and design processes may produce research and DTs that are considered more relevant by older adults (e.g., enhance face validity) and also ensures a bigger impact and challenges ageist assumptions [39]. More importantly, involving older adults is related to the human right of older adults to be involved in research and design that can potentially improve their quality of life [40]. Inclusion can promote older adults’ (including persons with cognitive decline) dignity, reduce the stigma and exclusion they endure in design, research, and society, and can give a sound to their unique voice [32]. Hellström, Nolan, Nordenfelt & Lundh further postulate that participation in studies can enhance a sense of worthiness, esteem and even have therapeutic effects for people with cognitive decline, contrary to the fear of researchers of causing emotional harm [26]. According to Hellström et al., [26], “the question is therefore not ‘should we?’ but ‘how can we?’” (p. 612) include older adults in research and design. However, how to do so remains a central challenge.

Therefore, the aim of this opinion paper is to advocate for the inclusion of older adults in research and design of DT and explain how to do so. Though guides and handbooks of designing and co-creation with older adults exist (e.g., [41,42]), it is still common that older adults are excluded or that their voice is not incorporated, both in the design process and research. We discuss research and design together, because these concepts are often inseparable in our view, as research (e.g., to assess needs) is a preliminary stage of design, whereas the design process often involves conducting research and using various tools to collect and analyze data in several stages of the development of prototypes. We also maintain that inclusion must be diverse and include older people from various age groups, gender, professional, economic, and cultural backgrounds. In this opinion paper, we elaborate on possible challenges and ethical considerations related to inclusion, as well as how to overcome them by providing general guidelines that can be followed and real life examples. Though perceived by some as difficult, inclusion is indeed possible and rewarding in our view. The intention of this opinion paper is to take the position that inclusion of older people in the research and design of DT is not only important but also necessary in designing DTs that match older adults’ wants and needs. Doing so will favorably contribute to the acceptance of these DTs.

Our goal is to persuade the reader that our position on this topic is a valid one. Therefore, we support our claims with practice-based evidence and published research studies. To do so, we reviewed the most recent literature, most of which was from the last decade. Our inquiry into the literature focused mainly on what is known about how DT is developed and studied in relation to older adults. We also focused our search on studies that discuss social exclusion, ageism and age stereotypes regarding DT. Previous experiences of the co-authors, as well as consultations and references from experts in the field of gerontology, gerontechnology and STS (science and technology studies) were used to complete our literature review. In the following section (Section 2), we further emphasize why it is fundamentally important to include older adults rather than exclude them in the research and design of DT in order to enhance use and acceptance, as well as to improve their quality of life. In Section 3, we provide practical considerations and guidelines for designers and researchers to include older adults (with and without cognitive decline) in practice, and we further demonstrate these guidelines with real life examples.

## 2. Why is it Important to Include Older Adults in Research and Design of DT?

The existence of a digital divide has been widely acknowledged over the years as representing a significant social and knowledge gap associated with the worldwide diffusion and adoption of technology [43]. This has been associated with a reported negative impact on health, income, civic participation and education [44]. Amongst the key socio-economic determinants of this divide is age [45,46], often referred to as the ‘grey divide;’ Evidence has pointed towards persistent self-exclusion by older adults, as well as exclusion by researchers, designers, marketing strategists and entrepreneurs involved in the realm of DT [47,48,49].

The access divide has received the most attention to date, and here access encompasses the complete course of adoption of technology from the possession of motivation and attitudes by users, to securing physical access to technology, to grasping digital skills, and eventually to attaining a frequency and diversity of usage of digital media [50]. In an attempt to deconstruct the homogenous representation of older adults’ as “non-users,” Quan-Haase et al. [51] found additional and more complex representations of older adults also as “Reluctants, Apprehensive Basic Users, Go-Getters, and Savvy Users,” (p. 1207). This was based on narratives that varied from the negative sense of digital media being overwhelming and a waste of their time to the positive sense that they can stay connected and learn new skills. Both the ‘Reluctants’ and the ‘Apprehensive Users’ have low digital skills, but the latter have slightly better engagement in online activities than the former. The ‘Basic Users’ and the ‘Go-Getters’ have mid-level digital skills, but the latter are more engaged in online activities. The ‘Savvy Users’ have high digital skills and are highly engaged in online activities. Rogers [43] cautioned that the access divide would be replaced by the learning divide, content divide, and by other types of divides through which technology would put adopters at an enormous advantage over non-users. The learning divide relates to one’s ability to learn how to use new DT and has been found to be influenced by age, social class, educational experience and geographical location. The content divide relates to the accessibility and clarity of text and content, with regards to people from different social and cultural backgrounds. Indeed, this is the case in the 21st century [52], whereby the acquisition of digital skills and competencies has moved from optional to essential. Therefore, advancing inclusion and digital engagement are requisite in order to deal with this issue. As the determination of age as a predictor of acceptance of DT is questionable [53], it is essential to understand that the reasons for non-engagement are more complex.

In this section, we want to further indicate the shortcomings of exclusion of older adults and emphasize why it is fundamentally important to include older persons in the research and design of DT. We posit that the exclusion of older adults is often affected by stereotypically viewing older adults as a homogeneous group whereby older people are often considered forgetful, more rigid in thought, less motivated, less dynamic than their younger counterparts, frail, ill, dependent and incompetent [19,54,55,56]. This exclusion can be considered as a form of ageism which covers biased knowledge, values, attitudes and behaviors towards older people [57]. Given the considerable discourse on demographic changes and the increase in the older adult population, one might think that DT researchers, designers and entrepreneurs might capitalize on the rising market size of people aged 65 years and above. However, the persistent reporting of the ‘grey digital divide’ suggests otherwise. Therefore, the shortcomings of the exclusion of older adults in DT are numerous.

First, there seems to be an apparent unexploited business opportunity to extend the focus to age groups (primarily the 80+ population) that are growing in numbers and needs and that have so far been neglected on the basis of assumptions that this age group is less capable of using or not interested in DT. There seems to be a lack of marketing research on older adults, as well as an anti-ageing trend in the expression of the marketing discourse [58]. Therefore, as the use of DT is being increasingly proposed in health and social care of older adults but designers of DT are still ill-equipped to design for this age group, engaging older adults as partners in design and research is a necessity to overcome these barriers [59].

Second and more importantly, there seems to be the missed opportunity for DT to substantially improve the quality of life of older adults who are living longer and who are experiencing multiple life changes and social isolation with the resultant negative consequences of depression and ill-health [3]. Several issues need to be considered, namely gaining a better understanding of factors that influence the acceptance of DT by older adults as a heterogeneous group, understanding their unique perspective that can be different than other stakeholders’ perspectives [28], understanding their motivations and needs [53] and gaining evidence of how DT can improve the lives of older people.

The systematic review by Peek et al. [13] is an eye-opener on why the ‘grey divide’ persists, as it exposes the overlooked complexity, multidimensionality and, to a certain extent, sensitivity of acceptance of technology for ageing in place. This review amply shows the intricacy that surrounds the diffusion and adoption of the use of DT in the older age group. Nevertheless, these factors are far from being unsurmountable. Indeed, there are examples of researchers and designers of DT who have addressed the sensory or cognitive decline of older adults during the various phases of adoption of technology from pre-implementation to post-implementation [13].

There is evidence of the use of simple to more sophisticated DT that can assist to meet some of the challenges older adults experience. For example, the use of information communication technology interventions and specifically of the internet as a communication tool has been associated with a lower level of loneliness [60,61], whereas the use of simple iPad technology has increased knowledge, elicited closer family ties, and led to a greater overall connection to society [3]. Similarly, Tsai et al. [62] found that observational and enactive learning played important roles for older adults in using tablets and overcoming technological self-inefficacy by seeing others using tablets, receiving support from family members, or being donated tablets. The outcome in this study was an increase in the sense of connectedness and digital inclusion among older adults. Pollack [63] provided more complex examples on how using artificial intelligence in assistive technology can assist older people with cognitive impairment to continue living in their homes by developing technology to support their functional independence. This is achieved by individually assessing cognitive status, compensating for impairment, and providing the assurance of safety or overseeing the performance of activities of daily living, depending on the degree of decline. Therefore, the inclusion rather than the exclusion of older adults and people with cognitive decline in research may provide a better and wider understanding of their variability in using DT. In the context of design, it may offer the identification of age-related short-comings of the intended designed technology.

The recognition of investing in older people and promoting them as active citizens and recipients of health and social care has also been receiving attention at state level. For example, the UK Government strategy for ageing in the 21st century has challenged conceptions of older people as passive by investing in a number of different projects [64]. One project explored the complex circumstances around which older people access and use social and community information and revealed the use of informal networks for securing information, advice and advocacy. The project showed that networks supported by technology-literate mediators, who may be older adults located within local community or voluntary organizations, are able to create digital ‘circles of support’ in order to generate and share self-authored content. Hence, not only can older adults be included in designing such services, they can also act as agents for enhancing wider inclusion.

It can be concluded that if DT is to fulfil the ‘promise’ of improving older adults’ well-being, it is necessary to comprehend the variability of older adults, the complexity of reasons that lead to their acceptance of DT, and their different needs. As is becoming clearer, age by itself cannot account for acceptance or inclusion [53]. Understanding the aforementioned issues can only be achieved by including older adults from different backgrounds in research and design that concern their needs.

## 3. How Can Older Adults Be Included in the Research and Design of DT?

Having established why it is important to include older adults, we now focus on the ‘how’ and suggest guidelines to promote inclusion in the research and design of DT. We further elaborate on the specific ethical aspects and considerations that researchers and technology designers might face in including older adults. We start by discussing awareness of stereotypes and ageism and how this might underlie design. We then elaborate on matters of obtaining consent across time and respecting autonomy. Finally, we address the use of tools and methodology, issues of privacy, safety and security (for a summary of considerations and guidelines, see Table 1), and we provide a demonstration of these guidelines using an example of actual co-designing with older adults (see Box 1).

Box 1Demonstration of ethical considerations and guidelines for inclusion of older adults in co-designing with older adults.In the following we describe unpublished findings from focus groups we have recently performed as part of a bigger study about co-designing. These focus groups were conducted with older adults who participated in co-design processes in an technological organization that provides entrepreneurs with a platform to try newly developed DTs in a living lab real life setting. During these focus groups, we asked participants about their experiences and impressions in these co-design sessions. Two examples described in the focus groups illustrate the topics mentioned in Table 1. One was described as a positive experience, whereas the second was perceived as negative.In the first example, participants described a positive experience of co-designing a social app that aimed to connect older adults with volunteering opportunities. Participants indicated that they were involved (voluntarily) in multiple sessions from the idea phase (before the app was even programed) through first versions of the app. They described feeling like “partners” in developing the app. Constant changes were made to the app after every iteration (session) according to the feedback from the older adults. Participants described how they gave feedback relating the usability and accessibility of the app (e.g., interface, font size, operation buttons, and icons), privacy and security issues, and they also gave ideas for how to disseminate the app and achieve higher acceptability. They further described how the idea behind the app and its feasibility increased between sessions.In the second example, participants described a negative experience of a single co-design session with a robot designed to assist the older adult. Participants described the setting as confusing and uncertain. They were asked to walk around the room with the robot following them. They indicated they did not feel they received sufficient instructions and information about their task. They further indicated that some features of the robot were problematic; for example, it walked behind them in what was intended to be assistance in carrying their shopping bags. Participants described this as stressful and unsecure, noting they always needed to look back. Furthermore, they indicated that the situation made them “feel old.” Some participants even indicated that it was too bad that the designers did not consult with them earlier (before making the prototype), as they could have given them “useful advice.” Participants also indicated that the designers should have included older adults who experience more physical difficulties than what they have, as they are more likely to be the ones who can benefit from such a device.These very different examples demonstrate the different aspects of the aforementioned guidelines. On one hand, a setting that activates stereotypes (feeling old and week), a low sense of knowledge, consent and control, and a general notion that they would not feel safe with this DT as it currently is. On the other hand, a co-design process where participants feel empowered and feel like partners, willingly consent to participate in multiple sessions, and with a high sense of confidence that this DT will meet their specific needs and expectations.

### 3.1. Awareness of Stereotypes and Ageism

As previously mentioned, technology can potentially improve care and well-being, and it can enable users to be connected to the outside world. Thus, the use of DT has consequences for older adults’ interaction with their environment. The appearance and aesthetics demonstrate how stereotypes can be embedded and perceived in the design of DT. Wearing or using assistive devices, e.g., a personal alarm system, can be a symbol of frailty and dependency and can therefore be considered stigmatizing [65]. It may “tag” certain groups, increasing the existing stigmatization towards them. For example, surveillance technologies might marginalize residents with dementia [66], and tracking devices could be embarrassing if they make noises in public [67].

Older adults should be included in developing these technologies to minimize the stigmatization and potential embarrassment caused by them. In developing a device, caution must be paid to size, weight and visibility to avoid stigmatization. In particular, older adults have been found to prefer to have the technology disguised as an everyday device [68]. Involving them in the research and design of devices can contribute information about how to design the device such that it will not stand out in oder to minimize the danger of avoiding social contacts due to feeling of stigmatization [69] or avoiding using the device [70]. Adults from different demographic backgrounds should be involved in the design of technologies, as the level of worry from stigmatization might differ among adults from different age groups, gender, cultural background and personality traits.

One solution for the danger of stigmatization is the adoption of a “universal design” which can be conceived as making more products usable by a wide range of people, not just older adults or people with disabilities. Such an approach can result in a more attractive design and products free from the stigma often associated with older adults [69].

### 3.2. Consent and Re-Consent

As mentioned earlier, concerns regarding older adults’ abilities to fully consent to participate in research (including design research), often drives exclusion. This is more prominent regarding older adults with cognitive impairments [27], but it is also due to stereotypically associating old age with cognitive decline [55]. Generally, consent procedures require patients to receive, remember, and comprehend information but also to appreciate the impact of the decision to participate [71]. Fully comprehending what a DT is doing in interventions and studies that involve technology should also consider the visibility (or non-visibility) of certain devices, such as in the case of sensors and monitoring technologies [36].

While a common notion would be that the natural process of ageing includes some degree of cognitive decline [72], studies have shown that when controlling for sensory decline, some cognitive abilities do not show decline [73,74] (see further details in Section 3.4). Furthermore, it seems that some sorts of self-regulation such as emotion regulation actually improve as we age [75,76]. The assessment of cognitive abilities might be challenging and is not always clear. One common approach could be assessing participants’ cognitive abilities, e.g., using the Mini Mental State Examination (MMSE) [77]. However, merely indicating or priming the possibility of cognitive ageing often reduces results of cognitive or memory testing [78]. Furthermore, despite scoring low on cognitive testing, some participants would be capable of demonstrating adequate understanding if researchers take the time to consider other options of explaining (e.g., multimedia) and extend the discussion about the study [79]. Some researchers have argued to not regard a person as incompetent until demonstrated otherwise [80], and some have also acknowledged that incompetence in one area does not automatically mean incompetence in others [81]. Hence, a broader and more holistic conceptualization of competence, beyond cognitive ability, seems justified and necessary.

How to communicate the goals of a study, the purpose of a certain DT, and the meaning of the process for the participant are therefore challenges. The use of corrective feedback, multiple learning trials, and the simplification of consent forms by cognitively adapting language may facilitate consent procedures [30,33,71]. There is evidence that simplified consent-forms involving figures or plain language increase understanding [71]. The timing and location of consent procedures also play a significant role. Choosing a place convenient for the person, with noise levels that are appropriate and in the time that best suits the person’s preferences are important to consent and participate.

### 3.3. Autonomy, Trust and Respect

Consent and willing participation also relate to the concept of autonomy. Autonomy concerns a person’s right to independently make their own decisions about their life. In relation to older adults, this is highly associated with decision making about care, housing, social activities and even the end of life. In today’s rapidly digitalizing environment, the implementation of DT use may assist older adults to live in their own home and facilitate the person’s engagement in everyday tasks [16]. In 2008, an AARP (American Association of Retired Persons) report showed that older adults are willing to use technologies that allow them to maintain social connections and independence [82]. The use of information and communication technology devices may also assist in communicating information and facilitating decision-making and access to services [83]. Opinions are divided as to how one’s health influences acceptance. A lower health status can be a barrier to using certain technologies but a facilitator to using others [84]. In a study by Pino et al. [85], it was found that older adults showed positive attitudes towards using heath-related technologies in the future, but only those with lower physical function would agree to use them in the present. Furthermore, older adults with lower physical function may agree more to adopt technologies that are considered more invasive, such as surveillance technologies [86]. Thus, assessing the person’s current needs and asking what they want are important to enhance acceptance.

The case of surveillance and monitoring technologies provides, on the one hand, the opportunity for older adults to stay in their home in a safe environment [87] while easing the concerns of family caregivers and professionals [88]. On the other hand, using such technologies raises ethical concerns regarding the older adult’s autonomy and involvement in the decision-making process to use these technologies [36]. That said, in the case of people with dementia who might wander, these technologies might enhance autonomy compared to more restricting alternatives [88]. Therefore, this issue raises many ethical questions for which the answer is somewhat inconclusive [66]. Implementing the use of sensors and surveillance technologies can only be achieved by assessing the person’s needs and wants at a particular time and place. It also must be embedded within the concept of person-centered care. If the benefits of technology are perceived to be favorable, then they must be evaluated to show that improvements as suggested by the older person have been taken into consideration. Furthermore, it is important to ensure that the older adult has the opportunity to withdraw from using the service [87].

Another example that has been receiving much attention is the use of social robotic technologies to provide person-centered interventions. Louie, McColl and Nejat [89] reviewed the acceptance and attitudes of older adults towards a human-like robot in order to determine if the robot’s human-like assistive and social characteristics would be acceptable to assist with activities of daily living. Overall, the research showed that older adults had a positive attitude towards the assistive robot and its applications. However, actual usage of such robots is currently limited, possibly because of high costs but also because the current user representation of a person using the robot is associated with weakness, frailty, and neediness [14]. It is important to recognize that up to one third of all assistive technologies are abandoned within one year of use, and for economic reasons, assistive technology developers do not always test the technologies and involve the people who may use it in the design process [90]. In many cases, older adults are involved in the design process of robots only after a prototype is already developed, and their views are not always considered by the designers [14]. Hence, involving older adults throughout the design process is important in order to customize appearance and to adapt how actions performed by the robots are adjusted to the needs of older adults (e.g., the speed of movement and processing time). Otherwise, these can turn out to be barriers to adoption [85].

Both examples demonstrate the importance of establishing trust and respecting the choices made by older adults. Trust and respect are strongly linked to the concept of autonomy. Achieving them lays in communicating what a study is about, what a specific DT does, what it will mean for the person using it, and what the person’s alternatives are. Such communication or interviews should take place in an environment where the person feels comfortable. It is important to explain and provide an opportunity to answer questions. In addition, it is important to have an interest in the person even after the “formal” part of the interview or design session and to avoid a “hit and run” approach. This is especially important in longitudinal settings where there is more than one measurement or in an ongoing design process.

### 3.4. Research Methods and Tools

Gaining consent to participate in design or research is merely an initial step. To better include older adults in the design and research of DT, it is important to have valid and adjusted research methods, tests and tools to collect data. Major tools for quantitative data collection may include paper- and computer-based questionnaire measurement and data collected by sensors such as wearable devices and GPS logs. The qualitative method usually includes in-depth interviews and focus group discussions. Some methods can be used in a mixed method approach, such as observations and self-report diaries. Currently, however, many tools and test materials are not ‘older adult friendly,’ and two main important aspects are neglected in designing the tools and test materials, namely the individual level and the societal level [91].

The first aspect is the individual level. In the natural process of ageing, older adults usually show different levels of sensory decline, especially auditory decline and visual decline, indicated by an increase in the use of sensory aids, such as hearing devices and glasses [92]. Such sensory decline can affect memory, information and cognition processing, and it might have an effect on research results and interpretation [93]. For example, a study among older adults aged 70–103 showed that 93.1% of the age-related variance in intelligence scores could be explained by visual and auditory acuity scores [94]. Taking that in account, Ben-David and others posited that if sensory decline is controlled or if the sensory context of the test is modified, then age-related changes in cognitive performance can be minimized or even effaced [73,74]. These findings suggest that cognitive decline, one of the main age stereotypes leading to exclusion, is not always the actual situation. Instead, accounting for sensory decline might enhance inclusion.

Therefore, we propose that to improve the inclusion of older adults in the use of tools and test materials, it is necessary to consider their sensory changes [91]. Specifically, in quantitative methods, tools such as questionnaires should use larger font sizes, enhance visual contrast, and consider other parameters such as increasing the light in the lab. In studies and design settings that involve the DTs themselves (such as GPS devices, wearables, or tablets), it is important to make sure that adequate instruction is provided on how to use the devices, charge, and maintain them. Moreover, pay attention that these do not cause discomfort or affect sensation levels according to the goal of the research. In qualitative methods, which mainly involve more bilateral interaction between the researcher and the participants, researchers should be aware to provide sound amplification, reduce background noise, and account for setting parameters such as time and location.

The second aspect is the societal level. Extensive research has postulated the impact of age-based stereotyping on test performance. Research has shown that older adults are commonly perceived as being dependent, senile and forgetful [78,95]. According to the stereotype embodiment theory, exposure to the negative stereotypes of old adults during one’s lifetime could lead to the implicit internalization of ageism. As people age, such internalized negative attitudes are directed towards their own age group and become a negative self-stereotype [96]. Various environmental stimuli can activate stereotypes and influence consequential behaviors [97]. Environmental stimuli can be blatant and explicitly remind people of a given stereotype—informing older adults before a cognitive performance task that “it is widely assumed that intellectual performance declines with age,” for example [98]. On the other hand, the stimuli can be subtle and presented in an implicit way. For example, filling in one’s age in the demographic information form can activate older adult’s self-stereotype [97]. Thus, it is advised that researchers should ask for demographic information at the end of the study.

Recognizing potential cues in designing research tools and test materials that may activate negative age stereotypes is therefore necessary and important [91]. Specifically, in quantitative methods, the instructions and verbal description in the research tools and test materials should avoid age-related cues such as ageist words or pictures that could activate age stereotypes among older adults. Especially in some experiment manipulations, research procedures should be carefully designed to avoid age stereotype activation. Likewise, in qualitative methods, researchers and DT designers should try to avoid age stereotypes and raise awareness of not being ageist in non-verbal interactions with older adults. Choosing the most suitable method is also of importance. Qualitative methods are often more suitable for research involving older adults with cognitive decline [26]. In the case of people with cognitive decline, try to avoid asking about ‘factual’ information (e.g., dates and names), as this might induce stress if the person does not remember. Instead, prefer to ask about experiences and feelings. If possible, it is recommended to ask permission to obtain factual data from family caregivers or professionals.

### 3.5. Privacy and Confidentiality

An additional aspect, often overlooked when involving older adults, relates to the invasiveness of DT or a service using DT. The use of technology should respect older adults’ privacy, especially when using it in private spaces such as homes or private spaces in nursing homes. This fear of harming one’s privacy is pronounced because the benefits of technologies in terms of health and safety can often overshadow rights to privacy [66].

Technologies can be used as a means of retaining autonomy and the quality of life for older people, including people with dementia, as they might enable people to continue to live in their own home. However, some technologies, such as surveillance technologies, can ultimately lead to a monitored and supervised life in the person’s home, resulting in a loss of privacy. This danger is particularly emphasized with real-time observation and with the use of cameras, particularly if the users cannot decide to turn them off [65]. As monitoring devices offer varying degrees of privacy loss, older adults should be included in their development to decide how much privacy loss is acceptable.

Furthermore, the inclusion of older adults can create better guidelines regarding the creation of an appropriate balance between needs and privacy. Participants can provide specific feedback regarding the conditions under which they are willing to compromise their sense of independence. It is possible that alternatives will be brought up, as certain devices will have lower degrees of privacy violations that will be more acceptable for participants, such as sensory systems with no cameras.

Older adults might also be more comfortable with a partial coverage of their house, such as in the kitchen but less so in their bedroom [99]. Their agreement might additionally depend on who is monitoring them (e.g., spouse or social services) [67]. People are more likely to agree to some degree of privacy violation if they believe that a balance has been struck between the benefits of monitoring and the perceived intrusion into privacy [100]. They are particularly likely to agree to some privacy loss from monitoring devices at their home due to their perception that these privacy losses might be much bigger if they would move to a nursing home [87].

### 3.6. Safety and Security

As mentioned in the previous section, DT involves information about the individual. Furthermore, DT in the context of older adults focuses mainly on care. Hence, an additional important aspect lays in ensuring that no harm will occur during the use of DT in interventions and the design of DT, and older adults using it should be provided with a sense of safety and security. Therefore, the research and development of such technologies should take the safety of their users into consideration. This can be done by explicitly addressing patient safety and by reporting the objective (e.g., adverse events) or subjective (e.g., participant sense of safety) measures of safety in such studies [101].

Including older adults in the process of co-design will allow for an understanding of the specific circumstances under which they will use the technology [102]. Their inclusion can increase the ecological validity of the safety measurement, as older adults can test and report on the safety aspects in settings that would be as similar as possible to the natural environment of the users, such as in their homes and without taking extra safety measures. This is in contrast to the more prevalent practice of studies reporting on safety, which are often conducted in a laboratory or community center setting and apply extra safety measures such as supervision or walking frames [103].

Older adults included in the development of technologies should represent different conditions and health status, as the (cap)ability of patients with different health needs can vary. Such inclusion will promote the adaptation of these technologies to different conditions and patients [104]. Moreover, due to the variability in the characteristics of potential users, adults from different backgrounds should be included. The inclusion of a variety of older adults will also allow for the identification of ways in which technologies can improve older people’s safety, security and ability to cope at home [102]. It will also enable the identification of domains in which their implementation can create a false sense of security, especially among staff in care facilities [66], and even create new risks, such as a delay in response by the staff [105].

In this section, we widely demonstrated the ‘how’ of including older adults in the design and research of DT. We provided many examples from the literature and practice on how involving older adults influences the design process, and we suggested means of doing so. We consequently believe there is abundant evidence to support our claim that the inclusion of older adults is both possible and recommendable.

## 4. Conclusions

The aim of this article was to advocate for the inclusion and involvement of older adults in the research and design of DT. We have described why it is important to include older persons, and we have suggested various means of doing so. Inclusion is important not only because it is just, but also because it can have an actual impact on the results of the design, research and acceptance of DT. As of today, there is a mismatch between the ‘promise’ of DT to improve older adults’ quality of life and the actual use of DT. Though some argue that the acceptance of DT and the digital divide are related to a generational gap and may improve in the future, current studies have shown this might not be the case, stressing that digital divide might even widen in the future [18,53]. Furthermore, as DT is exponentially changing and developing, the challenge of acceptance will probably continue to persist. Excluding and not taking into consideration what older adults want and think due to ageism can, in our opinion, be a barrier of acceptance. Thus, exclusion can, among other things, harm and defect the implementation of new DTs and research results. Awareness of this point and the means of how to do so are primary challenges. We have based these claims on ample evidence to demonstrate the negative effects of exclusion and, more importantly, the opportunity of including.

Negative stereotypes, ageism and the way older adults are involved in the design process and research of DT might play an important role in exclusion, consequently leading to the mismatch between what is currently designed and intended by designers and researchers for older adults and what they really intend to use. Older adults above the age of 65 are not a homogeneous group. Like all people, they have different backgrounds, capabilities, motivations and personalities. Hence, their exclusion from research or design based on chronological age is wrong. It is necessary to co-design with diverse groups of older adult stakeholders. Including considerations of function and disability, gender, professional and educational background, cultural differences, computer skills, and experience needs to be addressed in ascertaining the older person’s perception of using technology. Do not assume in advance that a person cannot participate just because they are old. Furthermore, as much of the designed technology focuses on care, more involvement of people with dementia and people who are experiencing complex health conditions is needed.

While exclusion in research and design can be one form of ageism, inclusion does not necessarily mean that research is conducted free of ageism. Specifically, it is not enough to co-design. It is also about “how” it is done. The setting in which the design or study takes place, the verbal and non-verbal communication of researchers and designers interacting with the older adults, and the study’s tools and methods can be influential. The way feedback from older adults is interpreted and accounted for, as well as the stage of the design process they are involved in, play an important role on the outcome. The possible connection of negative stereotypes and the expressions of ageism in the design process as possible barriers to using DT and ways to overcome it need further exploration.

This possible manifestation of ageism in the design process has not received much attention in previous publications. This opinion paper was based on a comprehensive literature review, but it was not systematic, which is its limitation. Therefore, future studies should try to systematically define, document and address ageism in the design and research process and its consequences. One focus should be on attitudes of DT designers, healthcare professionals, researchers and other stakeholders involved in the design process. Furthermore, it is also important to reflect on the possible role of internalized or embodied stereotypes [106] of the older adults themselves as a determinant of DT use and adoption.

This opinion paper demonstrated different methodologies and means of how to include older adults in a trustworthy and respectful way in research and design. As it was not our intention to cover all possible means and considerations of how to include older adults in design and research, our main and final remark is that it is important to think in an inclusive way. The responsibility for inclusion lays on all the stakeholders involved, including the older adults themselves, family members, ethic committees and policy makers. Designers and researchers should make the effort and dare to do it. It is important to ask the experts, namely the older adults. Furthermore, it is imperative to learn and seek advice from care professionals on how to communicate and how to overcome cognitive barriers if they exist and not to follow the easy path of not trying. Newly designed products, services or research projects will surely benefit from it.

## Figures and Tables

**Table 1 ijerph-16-03718-t001:** Summary of ethical considerations and guidelines for inclusion of older adults in research and design of digital technology.

Ethical Aspects and Considerations of DESIGNING and researching DT with Older Adults	Guidelines for Inclusion
Awareness of stereotypes and ageism	Pay attention to appearance and aesthetics. Older adults should be included in developing the external attributes of DTs to minimize possible stigmatization caused by them. Prefer disguising technology as an everyday device. Adoption of a “universal design” which can be conceived as making more products usable by a wide range of people, not just older adults or people with disabilities.
Consent and re-consent	Use a broader and more holistic conceptualization of competence beyond cognitive ability. Simplify consent forms by cognitively adapting language and using corrective feedback. Account for the setting. Choose a time and place convenient for the person with noise levels that are appropriate.
Autonomy, trust and respect	Assess the person’s needs and wants at the particular time and place. Provide an optional “exit” or possibility to withdraw using a specific DT (such as surveillance and monitoring technologies). Establish trust and respect the older person’s choices.
Research methods and tools	On the individual level, take into consideration and control for sensory decline. Adapt the use of fonts, contrast and visibility of materials. Furthermore, pay attention to sound amplification and reducing background noises. When using DT as part of a study or design, notice that adequate instruction is provided on how to use the devices, charge and maintain them. On the societal level, pay attention to possible cues (e.g., in trail instructions or setting) that can prime negative age stereotypes. Consider the most suitable method. Qualitative methods are often more suitable when involving older adults with cognitive decline. Prefer to ask about experiences and feelings rather than ‘factual’ information.
Privacy and confidentiality	For DTs that are invasive, older adults should be included in their development to decide how much privacy loss is acceptable. Provide control as to who has access to sensitive information about the older adult.
Safety and security	Prefer to design and study DTs in a natural environment so that issues of safety can be addressed. Include older adults with different conditions and health statuses in order to adapt and account for various situations.

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
