# Peer review of "Inclusion of Older Adults in the Research and Design of Digital Technology"

_ijerph, 2019, doi:10.3390/ijerph16193718_

Round 1

Reviewer 1 Report

Dear Authors,

This manuscript focuses on one of the challenges of our current society. The manuscript needs only minor revisions. There is the need for an additional section that expands on which are the future challenges and how they can be solved. An example that illustrates a real-case scenario needs to be included. First, you can list the issues; second, you can propose which could be the best way/ways to tackle them; third, explain how these solutions can positively impact future research and really improve the actual status of the art and make an impact in the near future (maybe also including a sketch, an image that resumes all the concepts). 

Minor Comments

Section 2, General Comment. Section 2 needs to be more critical. So far, it is a list of different things that are important to include older adults in research and design of DT but the reasons and critical points are not really addressed. It needs to be improved.

Section 2, lines 132-143. These concepts need to be clarified, described in a more concise fashion.

Reviewer 2 Report

I had the pleasure to review the Article “Inclusion of Older Adults in Research and Design of Digital Technology for the International Journal of Environmental Research and Public Health (IJERPH). The article deals with the important topic of the exclusion of older people from research and design of digital technologies and discusses ways to overcome these barriers. I really enjoyed reading the paper but nevertheless I have some structural and argumentative concerns and some minor inquiries for changes and therefore recommend to accept the paper with major revisions.

I have a few structural/argumentative concerns, which will be addressed in the following and a couple minor inquiries, which I will be discussed in the end.

                Structural/argumentative concerns:

A clear definition and systematization of what the authors mean by digital technologies throughout the article is missing. Especially since throughout the article the authors discuss various technologies (e.g. assistive, information and communication, monitoring) treating them equally in their arguments. It should be acknowledged though that different technologies might be more or less old-age-exclusive. Further it could be argued that not the technology itself, but their specific usage (e.g. tracking) results in ageist actions. Same goes for the relationship between research and design processes that needs some clarification. The majority of the article focuses on DT whereas only chapter 3.4 takes a closer look on research. It remains unclear if the authors included the discussion on research because they see it as part of the design process or treat it independently. If the latter was the aim the proportionality in the text between the two topics is off and needs to be adjusted. The authors focus mainly on the supply side, arguing that researchers and designers must be more aware of the potential exclusion of older people when designing DT. How such an exclusion relates to have or don’t have of technological competences on the side of older people is missing in the discussion. For example one could argue that as technology-affine generations age, the divide described by the authors could become less of a problem.

Concerns that need clarification:

It is not clear how the authors come up the literature that is presented and on which they base their arguments. Same goes for the presented summary of considerations and guidelines given in table 1. The results seem to be derived from literature presented earlier in the text, but it remains unclear what method was used (systematic lit-review, scoping-review)? Please clarify the methodological approach of the paper in general In the conclusion the authors state that their paper should be treated as an opinion paper. This should be made clear earlier in the text, and the results should be treated more carefully in regard to their generalizability and transferability.

Round 2

Reviewer 2 Report

The authors have successfully addressed all my major concerns regarding the paper. I therefore recommend to accept the paper in the present form.